# Distance-Dependent Distribution of Biomarkers in Colorectal Cancer Tissues: In Vivo Study

**DOI:** 10.3390/ijms26199367

**Published:** 2025-09-25

**Authors:** Tadeusz Sebzda, Jakub Karwacki, Mateusz Sobala, Henryk Filipowski, Mirosław Łątka, Jan Gnus, Jakub Gburek

**Affiliations:** 1Department of Pathophysiology, Wroclaw Medical University, 50-368 Wroclaw, Poland; tadeusz.sebzda@umw.edu.pl (T.S.); mateusz.sobala@student.umw.edu.pl (M.S.); speathf@gmail.com (H.F.); 2University Center of Excellence in Urology, Department of Minimally Invasive and Robotic Urology, Wroclaw Medical University, 50-556 Wroclaw, Poland; 3Department of Biomedical Engineering, Wroclaw University of Science and Technology, 50-370 Wroclaw, Poland; miroslaw.latka@pwr.edu.pl; 4Department of Physiotherapy, Wroclaw Medical University, 50-355 Wroclaw, Poland; jan.gnus@umw.edu.pl; 5Department of Pharmaceutical Biochemistry, Wroclaw Medical University, 50-556 Wroclaw, Poland; jakub.gburek@umw.edu.pl

**Keywords:** colorectal cancer, cathepsin B, cathepsin D, actin, in vivo study

## Abstract

Colorectal cancer (CRC) is among the most prevalent and deadly malignancies worldwide. Despite advancements in screening and treatment, its aggressive progression and tendency to metastasize remain major challenges. Biomarkers such as cathepsins B and D, actin isoforms, and cysteine protease inhibitors may influence tumor invasion and metastasis. However, little is known about their spatial distribution in tumor versus surrounding tissue. This study aimed to evaluate the location-dependent expression of selected biomarkers in CRC tissue to better understand their role in cancer progression. Tissue samples were obtained intraoperatively from 37 CRC patients at three locations: the tumor center, and 2 cm and 5 cm from the tumor margin. The activity and concentrations of cathepsins B and D, anti-papain activity, and actin fractions (globular [G-], filamentous [F-], and total actin [T-actin]) were measured using biochemical and spectrophotometric assays. Statistical analyses included ANOVA, MANOVA, and non-parametric tests, with significance set at *p* < 0.05. Cathepsin B activity was significantly elevated at the tumor center and decreased with distance from the tumor (*p* < 0.001). F-actin and T-actin levels followed a similar pattern, with significantly higher values near the tumor core (*p* < 0.05). Differences in G-actin were less pronounced. No significant spatial variation was found for cathepsin D, or anti-papain activity. The G-actin/T-actin and F-actin/G-actin ratios revealed significant shifts in actin polymerization states depending on the distance from the tumor. This study demonstrates spatial heterogeneity in the expression of key biomarkers in CRC tissues. Elevated levels of cathepsin B and altered actin dynamics in tumor regions suggest their involvement in local invasion and progression.

## 1. Introduction

Colorectal cancer (CRC) remains a significant cause of cancer-related deaths globally. Due to its high incidence and mortality rates, CRC poses a major public health challenge. According to the Globocan’s 2022 data, CRC ranked third worldwide for the number of new cancer cases, with 1,926,425 cases (9.6%), and second for cancer-related deaths, with 904,019 deaths (9.3%) [1]. The rising number of CRC cases can be attributed to factors such as an aging population, sedentary lifestyle, unhealthy dietary habits, and smoking [2]. CRC is characterized by a high mortality rate due to its rapid progression and metastasis [3].

The high mortality of CRC is driven by its aggressive progression and metastatic potential, which may be influenced by biomarkers such as cathepsins B and D, cystatin C, and actin. These molecules reshape the tumor microenvironment, facilitating invasion and dissemination. Cathepsin B, a protease that degrades the basement membrane, is overproduced by cancer cells, promoting angiogenesis and enhancing vascular endothelial growth factor (VEGF) and transforming growth factor-beta (TGF-β) release. Its increased secretion is linked to tumor invasion and metastasis [4]. Similarly, cathepsin D, active in the acidic tumor microenvironment, cooperates with cathepsin B in proteolysis, acting as a mitogenic factor that stimulates cancer growth [5,6,7]. Together, these proteases drive stroma degradation, enabling cancer cells to invade deeper tissues and migrate more efficiently. Actin, a key cytoskeletal protein, plays a crucial role in cell shape, adhesion, and division. In cancer, actin disorganization leads to reduced adhesion and increased motility, enhancing tumor invasiveness and metastasis [8]. Anti-papain activity reflects the function of cysteine protease inhibitors, which can act as reservoirs for cathepsins that may be released under pathological conditions [9]. Cystatin C is one of the primary extracellular inhibitors of cysteine proteases and has a particularly high affinity for cathepsin B, effectively suppressing its enzymatic activity [10].

The study aimed to explore potential correlations between the concentration of specific biomarkers and their distance from the tumor in CRC patients, with a focus on cathepsin B, cathepsin D, anti-papain activity, and actin (globular, filamentous, and total). This research provides insights into the biological characteristics of CRC, the effectiveness of current treatment approaches, and areas for improvement. Biomarkers play a crucial role in this context, offering valuable tools for early detection, prognosis, and monitoring of therapeutic responses.

## 2. Results

Among the 37 patients included in the study, there were 20 men and 17 women, with an average age of 68 years. The most common tumor locations were the rectum, affecting 12 patients, and the colon, found in 11 patients. Histological grading revealed that 31 patients had G2 tumors, while G3 tumors were observed in 6 patients. According to Dukes’ classification, the cancer was classified as stage B in 14 patients, stage C in 16 patients, and stage D in 7 patients (Table 1).

The median range of cathepsin B concentrations was 167.8–1274.2 mU/g at the tumor center, 50.6–550.5 mU/g at 2 cm from the tumor margin, and 18.9–515.1 mU/g at 5 cm from the tumor margin (Table 2) For globular actin (G-actin), the median values ranged from 76.0 to 181.0 U/mg at the tumor center, 76.8 to 169.0 U/mg at 2 cm from the tumor margin, and 60.0 to 162.0 U/mg at 5 cm from the tumor margin. Filamentous actin (F-actin) levels ranged from 670.0 to 2258.0 U/mg at the tumor center, 711.0 to 2288.0 U/mg at 2 cm from the tumor margin, and 263.0 to 2138.0 U/mg at 5 cm from the tumor margin. The values for total actin (T-actin) were 1001.0–2432.0 U/mg at the tumor center, 809.0–2301.0 U/mg at 2 cm from the tumor margin, and 345.0–2253.0 U/mg at 5 cm from the tumor margin.

Statistical analysis demonstrated significant differences (*p* < 0.05) in the levels of cathepsin B, G-actin, F-actin, and T-actin depending on the distance from the tumor (Table 3). For cathepsin B, the differences were statistically significant between the tumor center and 2 cm from the tumor margin (*p* < 0.001), between the tumor center and 5 cm from the tumor margin (*p* < 0.001), and between 2 cm from the tumor margin and 5 cm from the tumor margin (*p* < 0.001). For G-actin, a statistically significant difference was observed between the tumor center and 2 cm from the tumor margin (*p* = 0.049). The levels of F-actin were significantly different between the tumor center and 5 cm from the tumor margin (*p* < 0.001) and between 2 cm from the tumor margin and 5 cm from the tumor margin (*p* = 0.011). Similarly, for T-actin, significant differences were found between the tumor center and 2 cm from the tumor margin (*p* = 0.032), between the tumor center and 5 cm from the tumor margin (*p* < 0.001), and between 2 cm from the tumor margin and 5 cm from the tumor margin (*p* = 0.028). Figure 1 illustrates the distribution of biomarker concentrations relative to their distance from the tumor center.

## 3. Discussion

Our study investigated the concentrations and activity levels of specific biochemical parameters—namely, cathepsin B and D activities, anti-papain activity, protein concentration, and actin fractions (G-actin, F-actin, and T-actin)—in CRC tissues. Samples were collected from three distinct locations: the tumor center, 2 cm from the tumor margin, and 5 cm from the tumor margin. The findings provide insights into the biochemical landscape of CRC and its potential implications for tumor progression and metastasis.

Elevated cathepsin B activity was observed in tumor tissues compared to adjacent non-tumorous regions. This aligns with existing literature indicating that cathepsin B is upregulated in colorectal tumors and plays a role in tumor invasion and metastasis by degrading extracellular matrix components [11]. Similarly, increased cathepsin D activity has been implicated in cancer progression, with studies suggesting its involvement in activating other proteases that facilitate tumor invasion [12]. In a study conducted on 70 CRC patients and 20 healthy controls, cathepsin B levels were analyzed. The results showed that serum cathepsin B levels were significantly elevated in late-stage CRC patients with lymph node metastases compared to early-stage cases (2.9 vs. 0.33 ng/mL, *p* = 0.001). In 100 paired CRC tumor and adjacent normal tissue samples, cathepsin B protein expression was significantly higher in tumor cells compared to normal colonic tissue (*p* = 0.006), which aligns with the findings of our study [13]. Zhang et al. demonstrated that cathepsin B levels and the cathepsin B/cystatin C ratio were significantly elevated in CRC patients compared to healthy controls and individuals with benign digestive diseases (*p* < 0.01), suggesting their potential role in distinguishing malignant from non-malignant disease [14]. The study by Tamhane et al. found that cathepsin B levels were significantly increased in adenocarcinoma tissues compared to adjacent normal colon tissue and, notably, that cathepsin B localized not only to the extracellular space but also to the nucleus of mucosal cells in adenocarcinoma samples and in CRC cell lines. Their findings suggest the presence of a distinct 40 kDa truncated form of cathepsin B, which may influence tumor progression through mechanisms beyond extracellular matrix degradation [15]. Another study analyzing 185 CRC patients and 35 healthy controls found that cathepsin B levels were significantly higher in CRC patients compared to controls (16.1 ± 8.8 mU/L vs. 11.4 ± 6.5 mU/L, *p* < 0.05), reinforcing its potential as a biomarker for tumor detection [16]. However, the study did not find statistically significant differences in cathepsin B levels across Dukes’ stages, suggesting that while cathepsin B may be involved in CRC pathogenesis, its prognostic value in differentiating tumor stages remains uncertain. Moreover, another study found a significant association between cathepsin B levels and survival outcomes in CRC patients (*p* = 0.04), indicating its potential prognostic role. The Kaplan–Meier survival analysis demonstrated that higher cathepsin B concentrations correlated with poorer survival rates, supporting the hypothesis that cathepsin B contributes to tumor aggressiveness and metastatic potential [17]. Another study demonstrated elevated cathepsin B activity in tumor, liver, lung, and serum tissues prior to treatment, compared to negative control tissue homogenates. Cathepsin B and L activities were increased by up to 16-fold relative to control tissues before therapy, highlighting their potential role in tumor biology [18]. Another study found that cathepsin B activity was elevated by 26-fold in CRC tissue extracts compared to normal mucosa. Activity levels increased with the histological grade, from G1 to G3, and were higher in adenomas than in carcinomas. The study also measured cathepsin B activity in tissue extracts taken 2 and 5 cm from the tumor center, observing a gradual decrease in activity with distance, though values remained significantly higher than those in control tissues (*p* < 0.001) [19]. All aforementioned findings reinforce the diagnostic relevance of cathepsin B in CRC and align with our study’s observation of increased cathepsin B concentration in tumor tissues, further supporting its involvement in tumor progression and its potential as a biomarker for early CRC detection.

In terms of cathepsin D, a study on 110 Iraqi patients with colorectal tumors, including 90 cases of CRC and 20 benign tumors, investigated its expression using tissue microarray-immunohistochemistry (TMA-IHC). The results showed a significantly higher expression of cathepsin D in malignant tumors (96.7%) compared to adenomas (80%) and colitis cases (30%). Additionally, statistically significant differences (*p* < 0.001) were observed in cathepsin D expression across different tumor types (adenocarcinoma and mucinous carcinoma), tumor stages, and histological grades, highlighting its potential role as a marker of tumor progression in CRC [20]. Another study by Khalifa et al. demonstrated that cathepsin D was highly expressed in CRC tissues, with 90% of tumor cells and 92.5% of stromal cells showing positive immunostaining. Notably, cathepsin D expression in stromal cells correlated significantly with tumor invasion depth (*p* = 0.004), reinforcing its role in tumor progression and dissemination [21]. However, in our study, we did not observe the correlation between cathepsin D concentrations and the distance from the tumor center.

The analysis of actin fractions revealed alterations in the balance between G-actin and F-actin within tumor tissues. Actin remodeling is crucial for cell migration, invasion, and metastasis, and our findings align with studies showing cytoskeletal rearrangements in cancer cells. Ghazanfar et al. identified ACTBL2 overexpression in CRC tissues, suggesting actin isoform dysregulation contributes to tumor progression [22]. Similarly, Sousa-Squiavinato et al. demonstrated that cofilin-1 drives epithelial–mesenchymal transition (EMT) via actin cytoskeleton reorganization, reinforcing its role in CRC cell invasion [23]. Islam et al. further supported this by showing that inhibition of cofilin-mediated actin polymerization reduces CRC cell motility, confirming actin’s role in tumor cell migration [24]. Additionally, Gehren et al. linked actin cytoskeletal disorganization to weakened cell adhesion, facilitating cancer cell dissemination [25]. Collectively, these studies reinforce our findings that actin cytoskeletal alterations in CRC enhance tumor invasion, metastasis, and therapy resistance, making actin-targeting strategies a potential therapeutic approach [26].

Anti-papain activity, indicative of cysteine protease inhibition, was also assessed. Alterations in this activity may reflect changes in the regulation of proteolytic processes within the tumor microenvironment. Imbalances between proteases and their inhibitors can lead to increased degradation of extracellular matrices, promoting tumor invasion. Protein concentration measurements provided additional context to the biochemical environment of CRC tissues. Variations in protein levels between tumor and non-tumor regions may be associated with the metabolic demands of proliferating cancer cells and the remodeling of the tumor microenvironment. Another study examining anti-papain activity levels in human CRC tissue found that activity was significantly elevated compared to control tissue extracts (*p* < 0.001). However, no statistically significant differences were observed in relation to the distance from the tumor border, histological type, or tissue differentiation grade [19].

The limitations of this study include the relatively small sample size, which may limit the generalizability of the findings. Additionally, while this study provides important insights into the spatial distribution of biochemical parameters in CRC, further research is needed to determine the clinical significance of these variations. Future studies should consider incorporating larger cohorts and exploring how these biomarkers correlate with patient outcomes, including recurrence and survival rates. Another limitation is the lack of functional assays that could establish direct mechanistic links between observed biomarker variations and tumor behavior. Addressing these gaps in future research will enhance our understanding of CRC progression and improve the potential for biomarker-driven therapeutic strategies.

## 4. Materials and Methods

### 4.1. Patient Population

The study population consisted of 37 patients diagnosed with CRC, specifically adenocarcinoma, who were enrolled at the Provincial Specialist Hospital in Wroclaw. The analysis was conducted based on tumor location (colon, sigmoid colon, and rectum), tumor grading, and classification according to the Dukes’ staging system. The Dukes’ classification was defined as follows:Dukes A: Tumor invasion into the bowel wall without penetration beyond it.Dukes B: Tumor invasion through the bowel wall, extending into the muscle layer but without lymph node involvement.Dukes C: Lymph node involvement.Dukes D: Distant metastases.

Each patient underwent surgical intervention, and none received adjuvant therapy as part of their treatment regimen.

### 4.2. Biochemical Parameteres

Tissue samples were collected from all patients from the tumor center, as well as 2 cm and 5 cm from its margin intraoperatively. Parameters taken into consideration in the study comprised cathepsins B and D, anti-papain activity of the cystatin, total protein, and actins (total and according to the level of polymerization). Tissue homogenates preparation included homogenization in buffer A, composed of the following components:10 mM TRIS-HCl buffer (pH 7.4);0.1 mM adenosine 5′-triphosphate (ATP);1 mM dithiothreitol (DTT);0.1 mM CaCl_2_;0.25 M sucrose.

The tissue was homogenized using a Potter electric homogenizer for 2 min. The resulting homogenate was centrifuged at 105,000× *g* at 4 °C for 1 h. The obtained supernatant, representing the cytoplasmic components, was subsequently frozen at −70 °C. These prepared samples were thawed immediately prior to single-use biochemical analyses [27].

DNase activity was measured using a spectrophotometric micro-method based on the increase in absorbance at 260 nm, corresponding to the degradation products of highly polymerized DNA soluble in 15% HClO_4_. The assay conditions were optimized to detect nanogram quantities of DNase in the sample. The reaction mixture consisted of:0.2 mL of 0.2 M HEPES buffer (pH 7.0) containing 25 mM MgCl_2_ and 0.5 mM CaCl_2_;0.2 mL of bovine pancreatic DNase I solution;0.2 mL of a 0.1% aqueous solution of highly polymerized DNA;distilled water, added to adjust the final volume to 1 mL.

The reaction was conducted in a water bath at 37 °C and initiated by the simultaneous addition of DNA and DNase I solutions. After 15 min, the reaction was terminated by adding 0.5 mL of 15% HClO_4_, followed by incubation in an ice bath for 15 min. Undigested DNA was then removed by centrifugation at 15,000× *g* for 20 min. The absorbance of the clear supernatant was measured at 260 nm to quantify the degradation products of highly polymerized DNA [27].

The quantification of G-actin and T-actin was based on the inhibition of bovine pancreatic DNase I activity [28]. G-actin level was assessed by direct inhibition of DNase I activity, replacing distilled water in the reaction mixture with the tested supernatant. T-actin content was measured using a dilution test, in which samples were incubated in buffer A without sucrose (a buffer stabilizing G-actin). Each sample underwent a series of dilutions by gradually increasing the amount of buffer A while keeping the tested sample volume constant. The filamentous actin (F-actin) content was calculated as the difference between T-actin and G-actin levels. One unit of actin inhibitor was defined as the amount that reduced the activity of 20 mg of bovine pancreatic DNase I (used in the assay) by 10%. Actin concentration was expressed in DNase I inhibitor units per milligram of total protein in the sample.

Protein concentration was determined using the Lowry method, which involves spectrophotometric measurement of absorbance at 750 nm [29]. This method is based on the reduction of the Folin–Ciocalteu reagent by aromatic amino acids and copper complexed with peptide bonds. Cathepsin B levels and anti-papain activity were determined following the Barrett method [30,31]. Cathepsin D was measured spectrophotometrically at 280 nm after protein precipitation using trichloroacetic acid. Hemoglobin, denatured by acid, served as the protein substrate for cathepsin D.

### 4.3. Statistical Analysis

Continuous variables were described using the mean, standard deviation (SD), median, range, and sample size (*n*). Statistical analysis of continuous data included one-way analysis of variance (ANOVA) with Tukey’s post hoc tests and multivariate analysis of variance (MANOVA). For dependent samples, Student’s *t*-test was applied. For variables that did not follow a normal distribution, non-parametric tests were employed: the Mann–Whitney U test for independent samples and the Wilcoxon signed-rank test for dependent samples. Correlations between variables were assessed using Spearman’s rank correlation coefficient. The concentration and activity levels of specific parameters were presented in tables and illustrated using bar charts. A *p* value < 0.05 was considered statistically significant.

## 5. Conclusions

Our study demonstrates significant spatial variability in the concentrations and activity levels of key biochemical markers within CRC tissues. Cathepsins B and D exhibited elevated expression within tumor regions, supporting their involvement in local tissue invasion and metastatic potential. Additionally, alterations in actin cytoskeletal components, particularly increased concentrations of G-actin, F-actin, and T-actin at the tumor center compared to areas 2 cm and 5 cm away, highlight the role of cytoskeletal remodeling in cancer cell migration. These findings suggest a spatial correlation between tumor proximity and biomarker levels, reinforcing their relevance to tumor–stroma interactions. In contrast, no significant distance-dependent variation was observed for anti-papain activity or cathepsin D concentrations. These findings highlight the potential diagnostic and prognostic value of these biomarkers and emphasize the need for further studies to explore their application in CRC detection, prognosis, and treatment strategies.

## Figures and Tables

**Figure 1 ijms-26-09367-f001:**
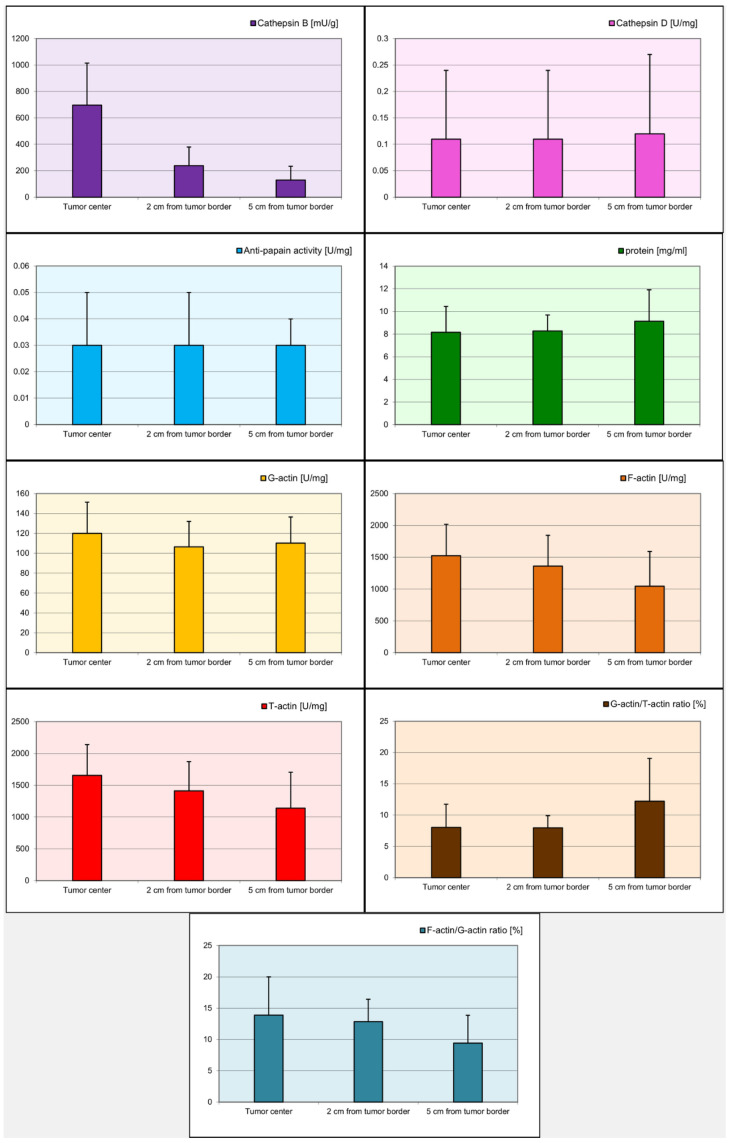
Visual representation of biomarker concentrations at varying distances from the tumor center. Corresponding *p* values are detailed in Table 1.

**Table 1 ijms-26-09367-t001:** Characteristics of 37 colorectal cancer patients.

Parameter	Study Population (*n* = 37)
Sex, *n* (%)	
Female	17 (45.9)
Male	20 (54.1)
Age, years	68.0 ± 5.6; 69.0 (59–78)
Tumor location, *n* (%)	
Colon	11 (29.7)
Sigmoid colon	7 (18.9)
Rectum	12 (32.4)
Any location + distant metastases	7 (18.9)
Histological grade, *n* (%)	
G2	31 (83.8)
G3	6 (16.2)
Dukes’ classification, *n* (%)	
B	14 (37.8)
C	16 (43.2)
D	7 (18.9)

Data is presented as mean ± standard deviation and median (range), or number of patients (percentage). Abbreviations: *n* = number of patients; G = grade.

**Table 2 ijms-26-09367-t002:** Concentration levels of selected biochemical parameters in colorectal cancer tissue samples, measured at three locations: the tumor center, 2 cm from the tumor margin, and 5 cm from the tumor margin.

Biochemical Parameter	Tumor Center	2 cm From the Margin	5 cm From the Margin
Cathepsin B, mU/g	696.2 ± 318.1; 654.1 (167.8–1274.2)	238.4 ± 141.2; 207.8 (50.6–550.5)	129.9 ± 103.9; 93.1 (18.9–515.1)
Cathepsin D, U/mg	0.11 ± 0.13; 0.05 (0.01–0.48)	0.11 ± 0.13; 0.05 (0.01–0.48)	0.12 ± 0.15; 0.04 (0.02–0.53)
Anti-papain activity, U/mg	0.03 ± 0.02; 0.02 (0.01–0.09)	0.03 ± 0.02; 0.03 (0.01–0.07)	0.03 ± 0.01; 0.03 (0.01–0.06)
Protein, mg/mL	8.2 ± 2.3; 7.8 (5.2–13.8)	8.3 ± 1.4; 8.5 (6.0–11.2)	9.14 ± 2.8; 8.5 (6.0–16.8)
G-actin, U/mg	119.8 ± 31.4; 128.0 (76.0–181.0)	106.3 ± 25.7; 95.8 (76.8–169.0)	110.2 ± 26.3; 112.0 (60.0–162.0)
F-actin, U/mg	1523.5 ± 492.4; 1445.0 (670.0–2258.0)	1362.0 ± 479.8; 1435.0 (711.0–2288.0)	1044.7 ± 546.2; 1115.0 (263.0–2138.0)
T-actin, U/mg	1655.4 ± 484.9; 1521.0 (1001.0–2432.0)	1411.84 ± 458.5; 1421.0 (809.0–2301.0)	1141.1 ± 563.6; 1212.0 (345.0–2253.0)
G-actin/T-actin ratio, %	8.1 ± 3.7	8.0 ± 1.9	12.2 ± 6.8
F-actin/G-actin ratio, %	13.9 ± 6.1	12.8 ± 3.6	9.4 ± 4.5

Data is presented as mean ± standard deviation and median (range) or as percentage. Abbreviations: m = milli-; U = unit; g = gram; L = liter; G-actin = globular actin; F-actin = filamentous actin; T-actin = total actin.

**Table 3 ijms-26-09367-t003:** Statistical significance (*p* values) of differences in biochemical parameter concentrations levels between different tissue sample locations in colorectal cancer patients. Comparisons were made between the tumor center vs. 2 cm from the tumor margin, tumor center vs. 5 cm from the tumor margin, and 2 cm from the margin vs. 5 cm from the margin.

Biochemical Parameter	Tumor Center vs. 2 cm From the Margin	Tumor Center vs. 5 cm From the Margin	2 cm From the Margin vs. 5 cm From the Margin
Cathepsin B	**<0.001**	**<0.001**	**<0.001**
Cathepsin D	0.283	0.242	0.250
Anti-papain activity	0.795	0.784	0.650
Protein	0.774	0.106	0.102
G-actin	**0.049**	0.164	0.524
F-actin	0.163	**<0.001**	**0.011**
T-actin	**0.032**	**<0.001**	**0.028**
G-actin/T-actin ratio, %	0.896	**0.002**	**0.001**
F-actin/G-actin ratio, %	0.381	**0.001**	**0.001**

Values in bold represent statistically significant correlations (*p* < 0.05). Abbreviations: G-actin = globular actin; F-actin = filamentous actin; T-actin = total actin.

## Data Availability

The data supporting this study’s findings are accessible upon request. Please contact the corresponding author to obtain the dataset.

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
