# Peer review of "Distance-Dependent Distribution of Biomarkers in Colorectal Cancer Tissues: In Vivo Study"

_ijms, 2025, doi:10.3390/ijms26199367_

Round 1
Reviewer 1 Report
Comments and Suggestions for Authors
The paper presents an interesting research on the different distribution of selected biomarkers in tumor tissue. Tumors isolated from 37 patients were analysed, in such a way that the expression/activity of selected biomarkers (cathepsin B and D, anti-papain activity, actin fractions and total proteins) was determined in the center of the tumor and in the surrounding area 2 and 5 cm from the tumor margin.
I have a few remarks about this work:
- I think that the title does not fully reflect what is presented in the paper, I suggest changing the title of the paper.
- It is not explained why total proteins are a biomarker of CRC, if they are not then they should not be mentioned in that context.
- I believe that the Introduction should be supplemented with the information provided in the Discussion in order to better justify the choice of biomarkers.
- In general, the claims stated in the Conclusions are not entirely based on the results presented in the paper.
These results lack a comparison of biomarker distribution with tumor grade or Duke grade to support the conclusions.
- Table 2 should be rearranged to make it clear which data belongs to which column.
In addition, the abbreviation for liter is L, not l, and the table and list of abbreviations should be corrected accordingly (l and ml should be changed to L and mL).
Figure 1 is completely redundant, contains no new information compared to Table 2, and is insufficient in terms of stating statistical significance.
Table 3 should be supplemented with a legend that contains an explanation of the meaning of the numbers written in bold (which means that statistical significance of <0.001 and <0.05 should be stated).
Author Response
The responses to the peer reviewers are in the attached file.

Reviewer 2 Report
Comments and Suggestions for Authors
The study by Sebzda et al. aimed to investigate the location-dependent expression of biomarkers such as cathepsins B and D, actin isoforms, and cysteine protease inhibitors in CRC tissue to better understand their role in cancer progression. This research may be suitable for publication in IJMC, but after some improvements.
Major recommendation:
In the introduction, explain in a few sentences the role of cystatin C and its relationship with anti-papain activity to explain the rationale for analyzing this biomarker in your study.
In the results, refer to tables and figures where appropriate.
In the table headings, the authors stated that they measured the concentration and activity of cathepsin B and D, but they only reported the data of the concentrations level presented as mean ± standard deviation and median (range).
What did the authors analyze from the preoperative blood serum samples, since they mention in Materials and Methods that blood were collected from all patients before surgery?
Since the increased cathepsin B concentration in tumor tissue is the most significant change found in this study, the authors should compare the differences in cathepsin B levels across Dukes’ stages, regardless of the relatively small sample size. Although study by Sebzda et al. found no statistically significant differences in cathepsin B levels across Dukes’ stages, the study by Abdulla et al. showed that serum cathepsin B levels were significantly elevated in late-stage CRC patients with lymph node metastases compared to early-stage cases, indicating a potential prognostic value of cathepsin B.
The authors should also analyze the association between cathepsin B levels and survival if they have survival data in their cohort.
In the Discussion section, in the paragraph discussing cathepsin D and its expression in CRC, the authors should emphasize that there are no statistically significant differences in cathepsin D levels depending on the distance to the tumor border. This should also be emphasized in the conclusion section.
Author Response
The response to peer-reviewers is in the attached file.

Round 2
Reviewer 1 Report
Comments and Suggestions for Authors
Dear authors, thank you very much for your response and the improvement of the manuscript.
Reviewer 2 Report
Comments and Suggestions for Authors
The authors addressed to all my suggestions and I recommend this paper for publication in IJMS in present form.